# Effects of Maitland Thoracic Joint Mobilization and Lumbar Stabilization Exercise on Diaphragm Thickness and Respiratory Function in Patients with a History of COVID-19

**DOI:** 10.3390/ijerph192417044

**Published:** 2022-12-19

**Authors:** Kyung-hun Kim, Dong-hoon Kim

**Affiliations:** 1Department of Physical Therapy, Gimcheon University, 214, Daehak-ro, Gimcheon 39528, Republic of Korea; 2Gimcheon Institute of Rehabilitation Science, Gimcheon University, Gimcheon 39528, Republic of Korea

**Keywords:** COVID-19, thoracic mobilization, lumbar stabilization, diaphragm, respiratory

## Abstract

**Objective**: We investigated the effects of maitland thoracic joint mobilization and lumbar stabilization exercise on diaphragm thickness and respiratory function in patients with a history of COVID-19. **Methods**: Thirty patients who had passed one month after COVID-19 onset were randomly divided into maitland thoracic mobilization and lumbar stabilization and combined breathing exercise groups; each group performed thoracic mobilization and lumbar stabilization exercises and combined breathing exercise and ergometer exercises, respectively, for 50 min, three times a week, for eight weeks. We used the MYSONO U5 MicroQuark to evaluate diaphragm thickness and respiratory function (forced vital capacity, forced expiratory volume in the one second, peak expiratory flow), respectively. **Results**: There were no significant between-group differences in general patient characteristics and change in diaphragm thickness and respiratory function. Both groups showed significant improvement within each parameter. However, the maitland thoracic mobilization and lumbar stabilization group showed more significant improvements than did the combined breathing exercise group (*p* < 0.05). **Conclusion**: In this study, we confirmed the maitland thoracic joint mobilization and lumbar stabilization exercise on the diaphragm thickness and respiratory function in patients with a history of COVID-19.

## 1. Introduction

The outbreak of coronavirus disease 2019 (COVID-19), caused by the new viral strain severe acute respiratory syndrome coronavirus 2 (SARS-CoV-2), occurred in China in December 2019, and a global pandemic ensued [1]. The clinical characteristics, comorbidities, and clinical outcomes of the disease were identified in each country around the world, from the United States and Europe to China [2,3,4].

The clinical symptoms of COVID-19 are nonspecific, and the clinical progression may start with an absence of symptoms to acute respiratory infection, severe pneumonia, and death. The common symptoms are fever and cough, as well as headache, sore throat, and muscle pain; severe symptoms include labored breathing, lethargy, fatigue, nausea, vomiting, diarrhea, and loss of smell or taste [5]. Approximately 80% of patients develop mild upper airway infection or mild to moderate pneumonia prior to recovery [6]; in some cases, mild symptoms may aggravate and progress to severe pneumonia or acute respiratory distress syndrome (ARDS). Signs of severe SARS-CoV-2 infection include labored breathing, hypoxia, or 50% subpulmonary defect on imaging in approximately 14% of patients. Life threatening events, ranging from respiratory failure to shock or multiple organ failure, occur in 5% of cases [7]. Moreover, for patients with acute conditions, survival is not the only goal, because COVID-19 survivors have been reported to experience persistent symptoms (fatigue, dyspnea, and chest pain) and frailty [8,9,10].

Joint mobilization is classified into various grades based on mobility [11]. As it can be applied to different parts of the body [12], joint mobilization may provide effective treatment of low-mobility joints and joints with limited range of motion (ROM) or functional rigidity [13]. Joint mobilization restores normal bodily functions by enhancing the performance of muscles surrounding the target joint [14]. Noll et al. [15] reported that joint mobilization facilitated physiological improvements, promoted autonomic nervous system response and lymphatic circulation through vertebral movements, and exerted functional effects to ensure vertebral posterior joint flexibility with thoracic cage mobility for enhanced cardiopulmonary function and reduced intensity of labored breathing.

Lumbar stabilization exercises improve pain and function in patients with back pain through core muscle activation [16]. Core stabilization muscles mainly comprise the inspiratory and expiratory muscles that promote spinal stability and postural control by increasing abdominal pressure [17]. In addition, lumbar stabilization exercises improve abdominal muscle activity in patients with stroke [18] and increase the transverse abdominis thickness in healthy individuals [19]. This may be related to respiratory function.

Previous studies confirmed the positive effects of joint mobilization and lumbar stabilization exercise on respiratory function; however, studies involving patients with a history of COVID-19 are lacking. Moreover, although studies have investigated persistent symptoms in patients who have recovered from COVID-19, the specific changes in respiratory function and the diaphragm as a respiratory muscle are yet to be adequately explored. Therefore, this study aimed to identify the effects of thoracic joint mobilization and lumbar stabilization exercises on diaphragm thickness and respiratory function in patients with a history of COVID-19 to provide a reference for interventions for improve respiratory function in such patients, including those with disabilities. The hypothesis of this study is that maitland thoracic joint mobilization and lumbar stabilization exercise will have a positive effect on diaphragm thickness and respiratory function in patients with a history of COVID-19.

## 2. Materials and Methods

This study was conducted based on a randomized, pretest-posttest controlled experimental design. A therapist with over five years of experience conducted the tests, analyses, and training using a single-blind design. The sample size was estimated by a statistical test using the G * power Version 3.1.9.7 (Franz et al., 2020) with an effect size of 0.68, significance level (α) of 0.05, and testing power of 0.8. The estimated sample size was 34.

### 2.1. Participants

The participants in this study were 34 patients at B Hospital, a general hospital located in Gyeonggi-do, South Korea, who voluntarily responded to the recruitment ad. The inclusion criteria were as follows: an individual who had passed 1 month after the onset of COVID-19 whose forced vital capacity (FVC) declined below 80% of the normal predicted value, and who underwent no breathing treatment, exhibited no cardiovascular disease or depression, whose Mini-Mental-State Examination-Korean (MMSE-K) score was ≥24 to indicate the ability to communicate and follow instructions, and who voluntarily provided written informed consent before initiation of the study. The exclusion criteria were as follows: an individual with a congenital or acquired deformity of the thoracic cage with a history of thoracic or abdominal surgery with an inability to perform respirator-based mechanisms and with orthopedic disease of the trunk [20]. This study was conducted in compliance with the Helsinki declaration. The Institutional Review Board (IRB) of Gimcheon University approved the study (approval number: GU-202207-HRa-06-03) (CRIS number: KCT0007806).

### 2.2. Protocol

Thirty participants were selected based on the inclusion criteria (four participants dropped out) and the objectives of the study. Participants were randomized between the thoracic mobilization and lumbar stabilization (TMLS, *n* = 15) and combined breathing exercise (CBE, *n* = 15) groups. To minimize selection bias, a digital random draw was used for the group selection.

The training program was applied for a period of 8 weeks, while pre- and post-tests were performed before and after the 8-week intervention, respectively. For the pre- and post-tests, rehabilitation ultrasound imaging was used to measure the variations in diaphragm thickness. In addition, the diagnostic spirometer was used to measure the FVC, forced expiratory volume in one second (FEV1), and peak expiratory flow (PEF) as indicators of respiratory function.

The 8-week intervention for both groups consisted of 24 training sessions, with three 50-min sessions conducted each week. The TMLS group performed thoracic mobilization and lumbar stabilization exercises for 30 min and lower limb ergometer exercises for 20 min. The CBE group performed combined breathing exercises for 30 min and lower limb ergometer exercises for 20 min.

#### 2.2.1. Maitland Thoracic Mobilization and Lumbar Stabilization Group

The TMLS group performed maitland thoracic mobilization and lumbar stabilization exercises for 30 min, followed by 20 min of lower limb ergometer exercise.

Thoracic mobilization was performed for 15 min using the joint mobilization method suggested by Maitland [11]. The grade of joint mobilization was determined based on pain and limitation in mobility. For Grade I mobilization, low-frequency vibration was applied from the starting point of the ROM. For Grades II and III mobilization, high-frequency vibration was applied from the midpoint and from the endpoint of the ROM, respectively. For Grade IV mobilization, low-frequency vibration was applied from the endpoint of the ROM. The patient was placed in a prone position and was instructed to adopt a slightly bent spinal posture, with the top section of the table being lowered. The therapist stood beside the patient with the lateral metacarpals or pisiforms on the thoracic vertebra of the patient. Subsequently, while standing immediately adjacent to the patient, the therapist stretched the patient′s arm with the shoulder positioned right above the spine, and then applied Grade II or III joint mobilization with weight transfer by hand (Maitland, 2005). To the vertebral segment with reduced mobility, mobilizations of the central vertebral postero-anterior-, transverse vertebral-, and unilateral postero-anterior joints were applied.

The lumbar stabilization exercises consisted of the following eight movements revised from those suggested by McGill & Karpowicz [21]: deep abdominal muscle strengthening, double-limb support bridge, single-limb support bridge, contralateral arm and leg abdominal lift, contralateral arm and leg press, leg press, contralateral arm lift, and unilateral arm and leg lift exercises with the participant in a prone position. Each movement was repeated three times. It was set as an intensity for maintaining posture. The duration of movement was 15 s, followed by 5–10 s of rest. Before and after the exercises, warm-up and cool-down were performed for 1 min each; the total exercise time was 15 min [22].

The lower limb ergometer exercises were aerobic exercises, performed for 20 min using a New 3000 (Shin Gwang, Republic of Korea) device. The exercise intensity was performed at a maximum heart rate (HR) of 40–50% for the first two weeks, 50–55% during the third week, and 55–60% for the fourth week. All participants wore an HR sensor to ensure that the exercise intensity was maintained [23].

#### 2.2.2. Combined Breathing Exercise Group

The CBE group performed 30 min of combined breathing exercise, followed by 20 min ergometer exercise three times per week for a period of 8 weeks.

The lower limb ergometer exercises were combined with diaphragm breathing and pursed-lip breathing exercises for the intervention. For the breathing exercise, the patient adopted a supine position and was instructed to perform inspiratory diaphragm breathing, followed immediately by expiratory pursed-lip breathing. The exercises were performed four times with a 1-min rest period after each session. Each session comprised five sets consisting of 4–5 breathing exercises with a 30s rest. The inspiratory diaphragm breathing exercise was performed as follows. The therapist placed a hand on the patient′s rectus abdominis right below the anterior costal cartilage, and the patient was instructed to inhale slowly and deeply through the nasal cavity to perform inspiratory diaphragm breathing. For this, the therapist applied a suitable level of resistance upon the rise of the rectus abdominis to induce deeper inhalation by the patient; the patient was instructed to relax the shoulders and maintain the relaxed state during deep inspiratory diaphragm breathing, while ensuring that the upper thorax remained immobile to only allow the abdomen to rise. The expiratory pursed-lip breathing exercise was performed after the patient inhaled deeply with pursed lips then breathed out the air for a set period of time.

The lower limb ergometer exercises were performed in the same manner as by the TMLS group.

### 2.3. Measurement

#### 2.3.1. Diaphragm Thickness

To measure changes in diaphragm thickness, ultrasound images were obtained using the MYSONO U5 (Samsung Medison, Seoul, Republic of Korea) for digital image analysis by rehabilitative ultrasound imaging (RUSI). The linear transducer (7.5 MHz), frequency modulation (6–8.5 MHz), and the range of gain (20–80) conditions were the same for all tests. First, the space between the eighth and ninth intercostal muscles along the right axillary line was marked with the patient in a supine position. Next, the instrument was moved vertical to the thoracic wall of the patient to measure the section of the diaphragm between the eighth and ninth intercostal muscles in the two-dimensional coronal plane. The process of maximum inspiration and expiration was repeated three times to facilitate the clear representation of the diaphragm on the screen. The thickness of the diaphragm during maximum expiration (at rest) and maximum inspiration (at contraction) was measured and the difference in the diaphragm thickness was estimated. The mean of triplicate measurements was obtained.

#### 2.3.2. Respiratory Function

For the pulmonary function test, the percent change-based spirometer MicroQuark (Cosmed, Italy) was used. For evaluation, the patient was placed in a sitting position on the bed. Each patient wore a nose clip to prevent the escape of air through the nasal cavity, wrapped the lips around the mouthpiece, and bit the mouth piece using their teeth. The FVC, FEV1, and PEF were measured, following the protocol for the maximal-effort expiratory spirogram. During the measurement, a mouthpiece was used on the part touching the mouth, which was immediately separated at the end of the measurement and alcohol-disinfected for hygiene control.

### 2.4. Statistical Analysis

PASW 20.0 for WINDOWS (SPSS Inc., Chicago, IL, USA) was used for all statistical analyses. The chi-square test was used to compare the sex, paretic side, and diagnosis among the general characteristics of the two groups, whereas age, height, weight, MMSE-K, and onset year were analyzed using independent t-test. Normality was tested using the Shapiro–Wilk test. The independent t-test was used to compare the variation by time before and six weeks after training between the two groups. The paired t-test was used for the difference in training result by time within each group. The significance level was set at α = 0.05.

## 3. Results

### 3.1. General Characteristics of Participants

Table 1 presents the homogeneity test results for the participants′ general and clinical characteristics.

### 3.2. Changes in Diaphragm Thickness Based on the Intervention Method

Table 2 presents the diaphragm thickness before and after the intervention in the TMLS and CBE groups. The measured dominant diaphragm rest increased significantly from 0.17 ± 0.05 cm to 0.19 ± 0.04 cm (*p* < 0.01) in the TMLS group and from 0.15 ± 0.03 cm to 0.16 ± 0.03 cm (*p* < 0.05) in the CBE group. A between-group comparison revealed that the variation was significantly higher in the TMLS group (0.02 ± 0.01 cm) than in the CBE group (0.01 ± 0.01 cm) (*p* < 0.05). The dominant diaphragm contraction increased significantly from 0.29 ± 0.06 cm to 0.39 ± 0.06 cm (*p* < 0.01) in the TMLS group and from 0.27 ± 0.03 cm to 0.32 ± 0.03 cm (*p* < 0.01) in the CBE group. A between-group comparison showed that the variation was significantly higher in the TMLS group (0.10 ± 0.02 cm) than in the CBE group (0.05 ± 0.02 cm) (*p* < 0.01). The DC was significantly increased (*p* < 0.01) from 73.07 ± 13.92% to 112.64 ± 26.02% in the TMLS group and from 79.84 ± 14.73% to 108.83 ± 29.29% in the CBE group, while the between-group comparison indicated no significant variation (*p* > 0.05).

### 3.3. Changes in Respiratory Function According to the Intervention Method

Table 3 presents the respiratory functions before and after intervention in the TMLS and CBE groups. The measured FVC significantly increased from 1.92 ± 0.50 L to 2.25 ± 0.39 L (*p* < 0.01) in the TMLS group and from 1.90 ± 0.50 L to 2.05 ± 0.44 L (*p* < 0.05) in the CBE group. Between-group comparison showed that the variation was significantly higher in the TMLS group (0.33 ± 0.25 L) than in the CBE group (0.14 ± 0.22 L) (*p* < 0.05). The FEV1 significantly increased from 1.66 ± 0.36 L to 2.09 ± 0.31 L (*p* < 0.01) in the TMLS group and from 1.62 ± 0.22 L to 1.88 ± 0.26 L (*p* < 0.01) in the CBE group. A between-group comparison revealed that the variation was significantly higher in the TMLS group (0.43 ± 0.19 L) than in the CBE group (0.25 ± 0.17 L) (*p* < 0.05). The PEF increased significantly from 2.75 ± 2.75 L to 3.15 ± 0.94 L (*p* < 0.01) in the TMLS group and from 2.62 ± 0.58 L to 2.81 ± 0.45 L (*p* < 0.01) in the CBE group. A between-group comparison showed that the variation was significantly higher in the TMLS group (0.40 ± 0.22 L) than in the CBE group (0.19 ± 0.17 L) (*p* < 0.05).

## 4. Discussion

Most patients are able to lead normal lives after SARS-CoV-2 infection, but certain symptoms may persist for several weeks or months in the acute phase after recovery [24]. Labored breathing as a COVID-19 aftereffect was reported at a rate of 24% [25]. In addition, the 2-month follow-up monitoring of patients with COVID-19 revealed that labored breathing and cough persisted in 53% and 34% of patients, respectively, while only 27% showed symptomatic improvements on the chest radiography [26], suggesting the need for intervention regarding respiratory functions in patients with COVID-19. Therefore, the study of intervention methods for improving respiratory function, as described herein, may be crucial and can markedly improve the rehabilitation of COVID-19, which is related to long-term sequelae.

Dean & Frownfelter [27] stated that the respiratory efficiency and mechanism could change according to the movement, damage, and asymmetry of the thoracic wall and the level of respiratory muscle paralysis. In order to resolve this, the study suggested that thoracic wall expansion and ventilation and the pulmonary capacity and volume should be adequately maintained. Watchie [28] suggested that thoracic and vertebral joint mobilization could address the problem of inefficient ventilation caused by cardiac pump dysfunction. Therefore, a critical factor in enhancing respiratory function is to improve the maximum contraction capacity of the abdominal muscles, which are closely associated with the FEV, as expiratory muscles, through thoracic mobilization and lumbar stabilization [29]. Therefore, this study determined the effects of maitland thoracic joint mobilization and lumbar stabilization exercises on diaphragm thickness and respiratory function in the intervention (*n* = 15) and control groups (*n* = 15), with combined breathing exercises in patients with a history of COVID-19. The results revealed significant improvements in both groups, while the between-group comparison indicated significantly greater improvements in the TMLS group.

A general goal of respiratory interventions is to restore the functions of the diaphragm after injury to prevent labored breathing and improve patients′ ability to exercise [30]. The results in this study showed that the diaphragm thickness varied significantly in both TMLS and CBE groups, but no significant difference was found in the between-group comparison. This agreed with the findings of Enright et al. [31], who reported morphological changes of the diaphragm with increased thickness and contraction upon maximum inspiration after an 8-week intervention of high-intensity inspiratory muscle training in healthy adults. Jung et al. [32] have reported that inspiratory muscle training improved diaphragm thickness in 29 patients with chronic stroke. Kaneko et al. [33] demonstrated that a change in diaphragm thickness upon maximum inspiration was closely associated with pulmonary capacity. The results suggested that increasing the diaphragm thickness could enhance the inspiratory muscle functions in association with physical working capacity and thereby improve exercise ability in healthy adults [31]. The change in diaphragm thickness in this study presumably had a positive effect on the diaphragm contraction with enhanced mobility of the thoracic joint, while the lumbar stabilization and ergometer exercises with active breathing facilitated the contraction of the core muscles, including the diaphragm. Thus, increasing the thickness of the diaphragm can be the basis for improving respiratory function.

The indicators FVC, FEV1, and PEF evaluated in this study are widely used indicators of respiratory function [13]. Our findings revealed significant changes in the FVC, FEV1, and PEF in both the TMLS and CBE groups, while the between-group comparison indicated a significant difference. Sutbeyaz et al. [34] evaluated the pulmonary function in patients with subacute stroke for 6 weeks. Interventions were applied in inspiratory muscle and respiratory retraining, and the main measures were compared to those of a control group; the inspiratory muscle training group displayed significantly increased levels of FEV1 and FVC. Park et al. [35] investigated the effects of supine spinal mobilization and lumbar stabilization exercises in 17 patients with back pain and showed significantly increased levels of FVC and FEV1, which conform to the results of this study. Additionally, Kim et al. [36] reported that a proprioceptive neuromuscular facilitation breathing exercise enhanced chest excursion and maximal respiratory pressure in 10 elderly subjects with inspiratory muscle weakness. As thoracic joint mobilization ensures that the respiratory muscle length is adequate for contraction and promotes muscle activities for improved ventilation [37], the thoracic joint mobilization and subsequent lumbar stabilization interventions in this study are presumed to have enhanced the respiratory function based on the stable activities of the respiratory muscles, including the diaphragm.

The main limitation in this study is the potential effect of daily activities on the dependent variables due to the lack of complete control of the participants′ daily lives. Furthermore, as this study involved a selected set of individuals who satisfied the inclusion criteria, care should be taken while generalizing the results regarding diaphragm thickness and respiratory function to all patients with a history of COVID-19. In the future, additional studies should be conducted involving a greater diversity of thoracic joint mobilization exercises; these should be based on the respiratory functions of individual patients with instruments to assess dependent variables.

## 5. Conclusions

This study adopted the methods used in previous studies to investigate the effects of maitland thoracic joint mobilization and lumbar stabilization exercises on diaphragm thickness and respiratory function in patients with a history of COVID-19. These exercises had positive effects on diaphragm thickness and respiratory function. Many patients with a history of COVID-19 experience respiratory dysfunction and COVID-19 aftereffects, which implies that functional recovery is a critical factor in the daily life of these patients. However, there is a lack of the relevant clinical practice evidence from patients with a history of COVID-19. Therefore, respiratory function training in patients with COVID-19, based on the intervention method used in this study, is recommended, and various additional studies should be conducted in the future. The significance of this study is that it assessed the intervention in patients with a history of COVID-19 who experienced prolonged aftereffects for an extended period.

## Figures and Tables

**Table 1 ijerph-19-17044-t001:** General characteristics of the participants (*n* = 30).

Parameters	TMLS Group (*n* = 15)	CBE Group (*n* = 15)	t/x^2^	*p*
Sex				
Male	10 (66.7%)	9 (60.0%)	0.144	0.71
Female	5 (33.3%)	6 (40.0%)		
Age (years)	54.20 ± 12.17 ^a^	53.20 ± 10.44	0.664	0.81
Height (cm)	166.60 ± 7.89	165.80 ± 6.54	0.171	0.77
Weight (kg)	61.97 ± 8.95	61.40 ± 8.50	0.403	0.86
MMSE-K	27.13 ± 1.60	27.47 ± 1.46	0.150	0.48

^a^ Mean ± standard deviation, TMLS: maitland thoracic mobilization and lumbar stabilization, CBE: Combined breathing exercise.

**Table 2 ijerph-19-17044-t002:** A comparison of diaphragmatic thickness pre- and post-intervention (*n* = 30).

Measures	TMLS Group (*n* = 15)			CBE Group (*n* = 15)			t	*p* ^(2)^
DR (cm)								
pre	0.17	±	0.05 ^(a)^	0.15	±	0.03	1.37	0.18
post	0.19	±	0.04	0.16	±	0.03		
Change	0.02	±	0.01	0.01	±	0.01	3.10	0.00 **
T (*p*) ^(1)^	6.61 (0.00 **)			2.55 (0.02 *)				
DC (cm)								
pre	0.29	±	0.06 ^(a)^	0.27	±	0.03	1.10	0.28
post	0.39	±	0.06	0.32	±	0.03		
Change ^(1)^	0.10	±	0.02	0.05	±	0.02	6.08	0.00 **
T (*p*)	16.44 (0.00 **)			9.86 (0.00 **)				
PC (%)								
pre	73.07	±	13.92 ^(a)^	79.84	±	14.73	1.29	0.21
post	112.64	±	26.02	108.83	±	29.29		
Change ^(1)^	39.57	±	19.98	28.99	±	199.51	1.47	0.15
T (*p*)	7.67 (0.00 **)			4.75 (0.00 **)				

(^a^) Mean ± standard deviation, TMLS group: maitland thoracic mobilization and lumbar stabilization group, CBE group: Combined Breathing Exercise group, DR: diaphragm rest, DC: diaphragm contraction, PC: percent change, (^1^) Paired *t*-test, (^2^) Independent *t*-test, * *p* < 0.05, ** *p* < 0.01.

**Table 3 ijerph-19-17044-t003:** A comparison of the respiratory function between pre-post (*n* = 30).

Measures	TMLS Group (*n* = 15)			CBE Group (*n* = 15)			t	*p* ^(2)^
FVC (L)								
pre	1.92	±	0.50 ^(a)^	1.90	±	0.50	0.09	0.93
post	2.25	±	0.39	2.05	±	0.44		
Change	0.33	±	0.25	0.14	±	0.22	2.19	0.04 *
T (*p*) ^(1)^	5.09 (0.00 **)			2.49 (0.03 *)				
FEV1 (L)								
pre	1.66	±	0.36 ^(a)^	1.62	±	0.22	0.30	0.77
post	2.09	±	0.31	1.88	±	0.26		
Change ^(1)^	0.43	±	0.19	0.25	±	0.17	2.72	0.01 *
T (*p*)	8.86 (0.00 **)			5.93 (0.00 **)				
PEF (L)								
pre	2.75	±	2.75 ^(a)^	2.62	±	0.58	0.45	0.65
post	3.15	±	0.94	2.81	±	0.45		
Change ^(1)^	0.40	±	0.22	0.19	±	0.17	2.94	0.01 *
T (*p*)	7.10 (0.00 **)			4.20 (0.00 **)				

(^a^) Mean ± standard deviation, TMLS group: maitland thoracic mobilization and lumbar stabilization group, CBE group: Combined Breathing Exercise group, FVC = forced vital capacity, FEV1 = forced expiratory volume in the one second, DDR: dominant diaphragm rest, DDC: dominant diaphragm contraction, PEF = peak expiratory flow. (^1^) Paired *t*-test, (^2^) Independent *t*-test, * *p* < 0.05, ** *p* < 0.01.

## Data Availability

Not applicable.

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
