# Peer review of "Effects of Maitland Thoracic Joint Mobilization and Lumbar Stabilization Exercise on Diaphragm Thickness and Respiratory Function in Patients with a History of COVID-19"

_ijerph, 2022, doi:10.3390/ijerph192417044_

Round 1

Reviewer 1 Report

Introduction:

- The significance of the study needs more explanation.

- The hypothesis of the study needs to be clarified.

Methods:

The stabilization exercise needs more details (frequency, mode, intensity, and duration).

Discussion:

The comparisons with previous studies need more details.

The strengths and implications should be discussed in detail.

Conclusions:

The recommendations and future directions need to be demonstrated.

Reviewer 2 Report

Dear Authors,

thank you for the opportunity to read this interesting study which aimed to assess the effects of scpecific physiotherapeutic interventions on respiratory function after COID infection. Unfortunatelly, manuscript cannot be published in this version. Please notice my main comments below:

1. Abstract: 

- abbreviations should not be used in abstract

- assessment methods were not cliarified

- material was not clarified - was it current infection, hom much time after infection?

- abstract is not free of errors, with lacking of scientific soundness

2. Introdunction:

- no justification of chosen methods - how  does lumbar stabilization  and joint mobilization influence on respiratory function?

3. Subjects

- FV<80% - is the measurement in the time of testing or during COVID inffection?

- what does ,,special treatment" mean?

in line 159 - in the text above there is information about upper limb ergometer, not lower limb

I have serious concernes about methodology of mobilization application - was it performed to all patients in the group? there is no information about contraindications for that kind of procedure, this technique should be only done in patients with joint hypomobility - what was NOT assessed according to Methods

Reviewer 3 Report

The study examined the effects of thoracic joint mobilization and lumbar stabilization exercises on diaphragm thickness and respiratory function in patients with a history of COVID-19 compared to a combination of breathing exercises and ergometer exercises. The work is well-designed and contains all the necessary parts. In summary, it should be emphasized that the patients had Covid and impaired respiratory function. It should also be clearly shown that the group with thoracic joint mobilization and umbilical stabilization exercises was also exercised on the ergometer.

Ambiguities about the number of patients should be explained. A sample of 34 patients was calculated, 30 were included in the study, and 35 were listed in the method.

Round 2

Reviewer 1 Report

All comments have been addressed. No further comments are required.